# Epidemiology and Economic Cost Analysis of Microbial Keratitis from a Tertiary Referral Hospital in Australia

**DOI:** 10.3390/pathogens12030413

**Published:** 2023-03-05

**Authors:** Jason Richard Daley, Matthew Kyu Lee, Xingdi Wang, Matin Ly, Chameen Samarawickrama

**Affiliations:** 1Liverpool Hospital, Sydney 2170, Australia; 2Faculty of Medicine and Health, The University of Sydney, Sydney 2006, Australia; 3Sydney and Sydney Eye Hospital, Sydney 2000, Australia; 4School of Medicine, The University of Notre Dame, Sydney 2010, Australia; 5Translational Ocular Research and Immunology Consortium (TORIC), Westmead Institute for Medical Research, Sydney 2145, Australia; 6Save Sight Institute, Central Clinical School, The University of Sydney, Sydney 2000, Australia

**Keywords:** microbial keratitis, epidemiology, microbiological profile, resistance patterns, corneal scrape, economic burden, admission, cost analysis, Australia

## Abstract

Microbial keratitis is the most common cause of infective vision loss. The causative organism varies by region, and most cases require intensive antimicrobial therapy. The purpose of this study was to analyse the causative organisms of microbial keratitis, its presentation and economic burden from a tertiary referral hospital in Australia. A retrospective review of 160 cases of microbial keratitis was performed, over a 5-year period from 2015–2020. A wide variety of costs were considered to determine the economic burden, using standardized data from the Independent Hospital Pricing Authority and the cost of personal income loss. Our study showed the most commonly occurring pathogens were Herpes Simplex (16%), *Staphylococcus aureus* (15.1%) and *Pseudomonas aeruginosa* (14.3%). A total of 59.3% of patients were admitted, with a median length of admission of 7 days. Median cost for all presentations of microbial keratitis was AUD 8013 (USD 5447), with costs significantly increasing with admission. The total annual cost of microbial keratitis within Australia is estimated to be AUD 13.58 million (USD 9.23 million). Our findings demonstrate that microbial keratitis represents a significant economic burden for eye-related diseases and the key driving factor for the cost is the length of admission. Minimizing the duration of admission, or opting for outpatient management where appropriate, would significantly reduce the cost of treatment for microbial keratitis.

## 1. Introduction

Microbial keratitis (MK) is a serious and sight-threatening ocular emergency that requires prompt diagnosis and intensive topical antimicrobial therapy to salvage vision [1,2]. It is caused by infection of the cornea by microorganisms, especially when the surface epithelium is compromised [3]. Risk factors for MK include contact lens use, ocular surface disease and trauma [1,2,3]. The cornea is especially vulnerable to microbial infection due to its avascular nature and poor visual outcome may result from a lack of aggressive, targeted antimicrobial therapy [4]. This allows deeper penetration of microbes with further corneal injury from both the pathogen and the host immune response, exacerbating permanent vision loss through scarring and vascularization [1,3,5]. Perforations of the cornea may arise within one to two days from particularly virulent infections [6,7]. As a subset of ocular surface infections, MK cases are more visually devastating, often require hospital admission for intensive topical antimicrobial therapy, and are associated with a greater economic burden to patients, healthcare and society [8].

Prompt diagnosis and treatment are crucial to the management of MK [9]. Corneal scrapes are performed to determine the causative organism and its antimicrobial susceptibilities [5]. The most frequently isolated organisms are *Staphylococcus*, *Streptococcus* and *Pseudomonas species*, though prevalence can depend on geographical regions and risk factors [10]. Severe contact lens-related infections tend to be Gram-negative, particularly *Pseudomonas species.* Other risk factors for keratitis include existing corneal or external eye disease, trauma, immune compromise, foreign body, and previous keratitis. Empirical treatment is based on the probability and likelihood of the causative organism in the geographical location. This emphasizes the crucial role that microbial surveillance plays in the treatment of MK and the importance of ongoing monitoring of resistance patterns to help establish effective, empirical antimicrobial therapy [4,11].

The incidence of MK varies greatly depending on location and population. It ranges from 6–52/100,000 in “Western” nations such as the United Kingdom (UK), the United States of America (USA) and Australia [12]. In south Asian countries, the incidence of MK has been reported to be as high as 113/100,000 in southern India and 799/100,000 in Nepal [13]. A significant proportion of these patients are admitted to hospital to assist in adherence to intensive topical antimicrobial therapy and enable closer monitoring. Cost evaluation of MK in the international literature is lacking with limited studies from the UK, USA, Taiwan and India [13,14,15,16,17]. The mean cost of procedural treatment from 2013–2018 for a patient with MK in the USA was estimated to be USD 1788.7 and in the UK as GBP 2855 (USD 3252) [14,16]. Due to the differences in healthcare systems and methodologies used, costs reported in previous studies vary widely. This study, based in Liverpool Hospital, an Australian tertiary referral centre, provides the most detailed economic cost analysis of MK to date.

The purpose of this study was to identify the range of causative organisms and their resistance patterns, along with the economic burden, prevalence, morbidity and cost analysis for MK in an Australian tertiary referral hospital. The study collated data on demographic characteristics, clinical features, economic burden, follow up, and visual outcomes of all MK cases in a sequential 5-year period from July 2015 to July 2020.

## 2. Materials and Methods

A retrospective review was conducted over 5 years from 2015 to 2020 of all MK cases at Liverpool Hospital, a tertiary referral ophthalmology unit consisting of sixteen ophthalmologists, one fellow and four trainees. All records of patients who were coded with MK related events by the Liverpool Hospital coding department were independently reviewed by three investigators. Coding events included: keratitis and keratoconjunctivitis, Keratitis unspecified, Keratoconjunctivitis, Corneal ulcer, Herpes viral keratitis and keratoconjunctivitis, Acanthamoebiasis, Other keratitis, Other superficial keratitis without conjunctivitis, Interstitial and deep keratitis and Herpes viral ocular disease.

The inclusion criteria for the study were all cases who required a diagnostic corneal culture (scrape or biopsy) processed by the local pathology network; smaller MK cases that were treated empirically were not included. Cultures were collected onto two glass slides for Gram stain, horse blood Columbia agar, chocolate Columbia agar and anaerobic blood agar and two enrichment slope mediums, composed of Sabouraud Dextrose Agar. They were inoculated and incubated at 35 °C in different conditions, then examined at 24 h and again at 48 h. Positive growth of a microbe and antimicrobial resistance was determined in accordance with the clinical breakpoint criteria used by the hospital laboratory, in accordance with local pathology guidelines. Bacteria were identified by matrix-assisted laser desorption ionization-time of flight (MALDI-TOF) and antibiotic susceptibilities were determined by an automated Vitek 2 system (bioMérieux, Inc., Marcy l’Etoile, France). Corneal scrapings were also taken for polymerase chain reaction (PCR) testing for the detection of herpes simplex virus, varicella zoster virus, fungi and Acanthamoeba DNA.

Basic demographic data such as age and gender, as well as other details of the medical history were also collected. The best corrected visual acuity (BCVA) was measured using Snellen projector charts, unaided or aided with their usual means of correction (glasses or contact lens). The Snellen fractions in combination with any letters incorrectly identified or any letters correctly identified on smaller lines were converted to the logarithm of the minimum angle of resolution (logMAR) for further analysis. Visual acuity for counting fingers was converted to 1.7 logMAR, hand movement to 2.0 logMAR, light perception to 2.3 logMAR and no light perception to 3.0 logMAR [18,19].

### 2.1. Cost Calculation

Costs were calculated for all presentations to Liverpool Hospital, diagnosed with MK. Costs were calculated for Accident and Emergency (A&E) presentation, hospital admission and outpatient follow up. Additional cost breakdown was performed for direct and indirect costs. Australian data from the Independent Hospital Pricing Authority (IHPA) was used as the basis for costing methods [20,21], and has been used in similar studies [22].

Direct costs were classified as those expenses directly relating to the delivery of patient care. Cost line items that contributed to these expenses included medical, nursing, allied health, pharmacy, critical care, operating room, use of special procedural suites, pathology, non-clinical services, and ward supplies. Indirect costs were classified as those supporting, non-patient related expenses and included the supporting aspects of the same cost line items, listed for the direct costs. Additionally, it also considered on-costs, hotel services and building depreciation [20,21]. This study also added lost patient earnings to the indirect costs, using the national median personal income from the Australian Bureau of Statistics [23,24].

For patients who were admitted to hospital, diagnosis-related group (DRG) codes, from the clinical information department at Liverpool Hospital were used. For patients not admitted, costs were calculated using the Australian Emergency Care Classification (AECC) [25,26]. Public outpatients follow up was costed using the standardized price listed for an ophthalmology outpatient clinic by the IHPA [21,27,28]. This was a uniform cost of AUD 297 (USD 202) per clinic, including the direct and indirect costs, but not including the lost wages of the patient. Private room follow up was costed as AUD 78.05 (USD 53.06) [29]; however, no further data were able to be collected from the private ophthalmology practices, and therefore all possible additional costs associated with private follow up have been omitted.

This study also attributed personal wage loss of patients as indirect costs. This was calculated by multiplying the total number of days lost by AUD 161 (USD 110), the daily median wage in Australia in 2021 [22,23,30]. The total number of days lost was defined as the sum of the length of admission OR if the patient was not admitted, the day of A&E presentation, plus the total number of clinic visits. It was assumed that there was a one-day loss for each follow up appointment in the outpatient clinic. We did not include any days that patients could not attend work but were not directly associated with a clinic visit.

### 2.2. Statistical Analysis

Statistical analysis was performed using IBM SPSS statistics desktop version 28 (IBM Corporation, Armonk, NY, USA). The summary statistics were presented by number (percentage), mean (SD) or median (IQR), as appropriate. To explore the associations between various clinical factors and length of admission plus total costs per presentation, we used linear regression models, adjusted for age and sex. A Chi-square test was also conducted to assess the effect of topical steroids on median epithelial defect resolution time and scar formation. *p* < 0.05 was regarded as statistically significant.

## 3. Results

### 3.1. Demographics

There were 160 cases of MK that presented to Liverpool Hospital between July 2015 and July 2020. Table 1 presents demographics, medical history, risk factors and slit lamp examination findings of our cohort.

### 3.2. Microbiological Profile

The culture positivity rate was 54.1%. Of the culture positive cases, the majority were bacterial, constituting 75.2% of cases, followed by viral 19.3%, fungal 3.7% and acanthamoeba 1.8%. There were 10 cases of polymicrobial infection. Of all organisms identified on corneal scrape, Herpes Simplex Virus was the most common, making up 16%, followed by *Staphylococcus aureus* with 15%, then *Pseudomonas aeruginosa* with 14% and *Moraxella species* with 12% (Table 2).

The antimicrobial resistance patterns for the most commonly isolated organisms have been summarized in Table 3. Our cohort demonstrated a high sensitivity to fluoroquinolones. All five isolates of MRSA were resistant to fluroquinolones but were sensitive to vancomycin. *Streptococcal species* were not tested for fluoroquinolone sensitivity (due to laboratory protocols) but were sensitive to penicillin. The susceptibility of organisms to cefazolin was extrapolated from testing penicillin.

### 3.3. Clinical Outcomes

The most common, empirical antibiotic regimen used was ofloxacin monotherapy (n = 88, 55% of cases), followed by fortified cefazolin 5% and gentamicin 0.9% drops (n = 33, 33%). The most common ocular specific adjunctive medications included topical atropine (n = 44, 27.5%), oral valaciclovir (n = 33, 20.6%) and oral doxycycline (n = 31, 19.4%). A total of 50 patients did not receive any adjunctive treatment. Topical steroids were prescribed in half the cases of MK (n = 79/160) and the median time to their implementation was 4 days (IQR 2–7). The main topical steroids prescribed were dexamethasone (n = 31, 19.4%), prednisolone sodium phosphate minims (n = 29, 18.1%) and prednisolone acetate (n = 25, 15.6%). In the steroid group, the median visual acuity at initial presentation in the affected eye was 1.00 logMAR (20/200) compared to 0.69 logMAR (~20/100) in the non-steroid group. However, at final visit, the steroid group visual acuity was 0.45 logMAR (20/56) compared to 0.48 logMAR (20/60). The non-steroid group had a lower likelihood of developing a scar, 32% versus 57% (*p* = 0.001). Both groups had the same median length of admission of 7 days.

Admission was associated with age and severity of infection (initial visual acuity and size of epithelial defect (ED)). The likelihood of admission increased by 2% for every year of age (OR, 95% CI: 1.02, 1.01–1.04, *p* = 0.001). An initial BCVA in the affected eye of worse than 0.2 logMAR (20/40) was associated with a 3.6-fold increase in the likelihood of admission (OR, 95% CI: 3.55, 1.69–7.47, *p* = 0.001). Similarly, for every 1 mm^2^ increase in the area of the epithelial defect there was an 11% increase in the likelihood of admission (OR, 95% CI 1.11, 1.04–1.18, *p* = 0.001). There was no statistically significant relationship between the likelihood of admission and comorbidities such as cardiovascular disease, diabetes, autoimmune conditions, or malignancy.

There was a significant relationship between area of the epithelial defect and length of admission, where an ED of greater than 6.92 mm^2^ (β co-efficient, 95% CI: 6.92, 2.29–11.54, *p* = 0.004) was associated with longer admissions (≥6 days). Hypopyon was seen in 44 cases (27.5%). Corneal thinning was present in 61 (38.1%) of eyes, and of these eyes, 12 (19.7%) had a corneal perforation. The median epithelial defect healing time was 15 days. A definitive corneal scar was present in 43.8% (n = 70) of patients at their final clinic visit. Surgical intervention was required in 5.6% (n = 9) of patients. The presence of hypopyon, corneal thinning or perforation, did not have a statistically significant impact on length of admission. With regard to final BCVA the outpatient group had 56% better vision in logMAR than those treated as inpatients (OR 0.44; 95% CI: 0.44, 0.28–0.71, *p* = 0.001).

### 3.4. Economic Burden

There were a total of 1398 outpatient follow up appointments in the public ophthalmology clinic. The median number of outpatient clinic follow ups was 7 (IQR 3–17) for admitted patients, and 5 (IQR 1–9) for those treated as outpatients. There were 21 patients who also followed up in private rooms but were lost to this study’s follow up after their first consultation. The total number of patient days lost (i.e., the number of days the patient would be unable to attend work due to admission or presentation to A&E or clinic) was 2890, with a median of 16 days (IQR 9–27) for admitted patients and 6 days (IQR 3–10) for patients treated as outpatients only.

Costs were determined for admission, treatment in A&E and follow up in outpatient clinics (Figure 1). These costs were further broken down into direct, indirect and total costs, with wages lost contributing to indirect costs (Table 4, Figure 2). The cost of admission was by far the most significant contributor, accounting for 71% of the costs. Advanced age was found to be a statistically significant factor in cost, increasing by an average of AUD 250 (USD 170) per year of life (β co-efficient, 95% CI: 250, 91.8–407.3, *p* = 0.002).

Additionally, this study found that the total annual cost of MK at Liverpool Hospital was approximately AUD 581,748 (USD 395,472). Using an estimated incidence of MK as 0.66 cases per 10,000 people [12] and a current Australian population of 25.69 million, we estimated that there are approximately 1695 new cases of MK annually in Australia. With the median cost of an MK presentation at AUD 8013 (USD 5447), combining all presentations in both inpatient and outpatient settings, we calculated the total annual cost of MK in Australia to be approximately AUD 13.58 million (USD 9.23 million).

### 3.5. Risk Factors

The relationship between final BCVA and MK risk factors was evaluated. The most common risk factors for MK were existing corneal disease (n = 25, 15.6%), previous keratitis (n = 24, 15%) and external eye disease (n = 24, 15%). Due to the older patient demographics, contact lens use (12.5%) was the fourth most common risk factor. Linear regression revealed that previous keratitis was the only risk factor that was associated with a worsening of final BCVA, by 1.1 logMAR (β co-efficient, 95% CI: 1.1, 0.7–1.6, *p* < 0.001).

## 4. Discussion

Amongst the 160 cases of MK who were managed at Liverpool Hospital from 2015–2020, there was a culture positivity rate of 54.1%. The causative organisms were bacterial in 75.2%, viral in 19.3%, fungal in 3.7% and acanthamoeba in 1.8%, with 9.2% of cases being polymicrobial. These results are comparable to similar surveillance studies completed worldwide [8,31,32]. Antibiotic sensitivity results were available for 75 of the 94 (79.8%) bacterial isolates. Overall, a high fluroquinolone sensitivity was recorded with 29/30 (96.7%) of bacteria being sensitive. Of note, *Pseudomonas* demonstrated excellent sensitivity to fluoroquinolones. This is an important finding, as empirical antibiotic protocols can broadly be divided into two strategies—either fluoroquinolone monotherapy or dual therapy with fortified aminoglycoside and cephalosporins. Our data demonstrated increasing resistance of Gram-positive isolates to cefazolin, with a 40% resistance rate. This was also reported by Mun et al. [33] who found a 46% resistance rate to cefazolin and Leibovitch et al. [34], 34% to cefazolin. An increasing concern has also been fluoroquinolone resistance in Pseudomonas strains reported in the international literature, decreasing the effectiveness of empirical monotherapy [33,35,36]. However, this was not our finding. Indeed, low rates of fluoroquinolone resistance have been repeatedly documented within Australia [8,11,37], with a major factor being their restricted access, limiting their inappropriate use [38]. Overall, we advocate for fluoroquinolone monotherapy as the preferred empirical treatment for bacterial keratitis in our region, as beyond the low levels of resistance, there are benefits in improved patient compliance, commercial availability in a stable and preserved formulation and cost effectiveness as it negates the need for specialist compounding pharmacies to formulate the fortified antibiotics.

The role of steroids in MK was defined in the SCUT trial [39] and has been expanded and added to over the last decade [40,41,42,43,44]. It is postulated that steroids act by minimizing macrophage activity, thus reducing host mediated destruction of the cornea even though it delays re-epithelization [39]. However, many clinicians are still nervous about using steroids and avoid their use, for fear of exacerbating MK. In our cohort, topical steroids were used in about half of the bacterial keratitis cases, with generally more severe cases receiving steroids. Though the steroid group had worse vision at presentation, their final visual acuity was better than those who did not receive any steroids. These results were surprising, though perhaps not unexpected. We anticipated that the milder infections who did not receive steroids, would maintain their superior BCVA over the more severe infections that did receive steroids. However, this was not demonstrated in our cohort, and our data adds to the growing body of literature that advocates strongly for the use of steroids where appropriate, to maximize the visual recovery in patients with bacterial keratitis.

MK constitutes a significant total cost to the individual and health care system. The last costing study for MK in Australia used data from 2003 and found that the total costs ranged from AUD 8151 (USD 5541) for severe cases with vision loss and AUD 1175 (USD 799) for mild MK (adjusted for inflation in 2021) [45]. To improve the accuracy and generalizability of our estimation, we used IHPA data, which collates costs, for specific conditions, from over 500 hospitals around Australia. IHPA data uses the same costing foundations that fund hospitals for each episode of MK, making the data more consistent and translatable across other public hospitals. Further, we included a wider variety of costs not considered in other studies, such as allied health, ward supplies, non-clinical staff and services [16,45,46]. We estimated the total annual cost to Australia to be approximately AUD 13.58 million (USD 9.23 million) with a median cost per patient presentation of AUD 8013 (USD 5447), similar to more recent reports from the UK [16]. However, the greatest contributor to this cost was hospital admission, which had a median cost of AUD 11,998 (USD 8159) and total of AUD 18,003 (USD 12,242), when including potential wages loss and follow up post discharge. This is in comparison to those treated purely in an outpatient setting whose median cost was less than 25% of those for an admitted patient. From a health economics perspective, given the relative underfunding of most public health systems, it is important to try and minimize the cost of managing diseases. In the context of MK, a 75% saving can be achieved by managing patients without admission, where appropriate, and this represents a significant reduction in costs without impacting the health outcomes for the individual.

There were some limitations with our study. As a retrospective review this study is subject to confounding as it is unable to control some variables from the outset. We only included the results of MK that warranted a corneal scrape and could not include those whose keratitis was small and resolved with standard empirical treatments. Antibiotic sensitivities were based on systemic breakpoints and does not reflect the suprathreshold concentrations that can be achieved with topical administration. Our costing estimates were limited by not having access to private ophthalmology records. Thus, we do not have records of the number and frequency of follow-up, and instead gave a single estimate of AUD 78.05 based on the Medicare rebate in Australia. Equally, we did not have records for the number of days a patient did not attend work and instead calculated the number of clinic visits and costed that based on the median income. Beyond this we have no method to determine the psychological impact on the individual, the impact on family and friends who support the patient or the impact on their career. Taken together, it is likely that we have underestimated the true cost of keratitis by order of magnitudes.

## 5. Conclusions

MK represents a significant economic burden to the healthcare system, costing approximately AUD 13.58 million/year in Australia (USD 9.23 million/year). Monotherapy with a fluoroquinolone was found to be the most effective empirical therapy for MK in our region, with minimal evidence of fluoroquinolone resistance. Topical steroids were found to improve visual outcomes and can be safely used in both mild and more severe bacterial infections, where clinically appropriate. While early and intensive treatment is essential for sight preservation, extended patient admissions will disproportionately increase the financial burden on the healthcare system, compared to outpatient treatment. Therefore, where appropriate, clinicians should treat MK in the outpatient setting and in cases where admission is required, shift care to an outpatient basis as quickly and safely as possible.

## Figures and Tables

**Figure 1 pathogens-12-00413-f001:**
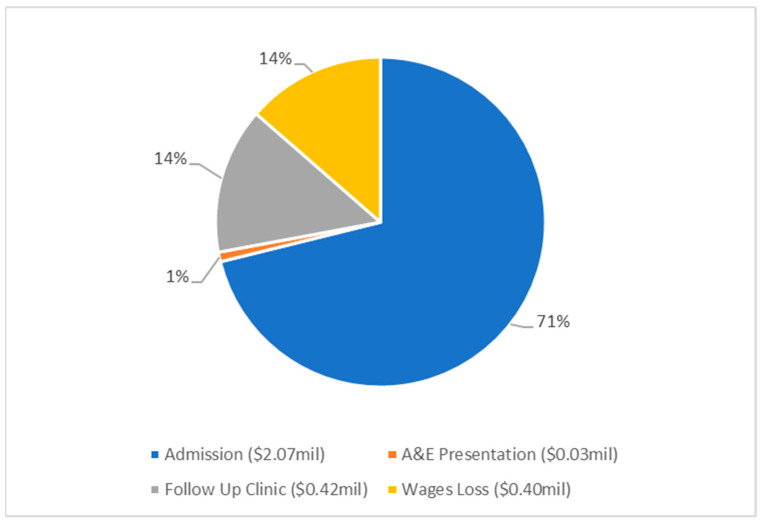
Pie chart presenting the economic cost analysis for MK, at Liverpool Hospital from 2015–2020, totalling AUD 2.91 million (USD 1.98 million) according to key categories of expenditure. All costs are shown in Australian dollars.

**Figure 2 pathogens-12-00413-f002:**
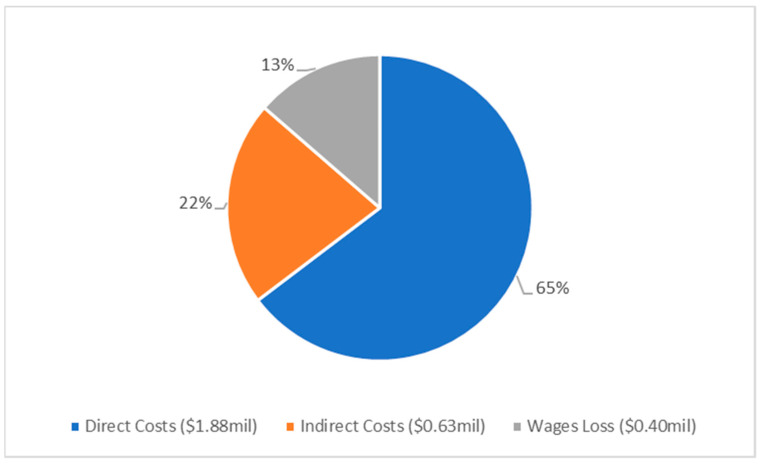
Pie chart presenting the analysis of direct and indirect costs of MK, at Liverpool Hospital from 2015–2020, totalling AUD 2.91 million (USD 1.98 million). All costs are shown in Australian dollars.

**Table 1 pathogens-12-00413-t001:** Demographics, clinical features, and risk factors for MK patients.

**Demographics**	
Age (mean ± SD, range years)	57.6 ± 24.6, range 5–101
Female, n (%)	68 (42.5%)
Admission, n (%)	95 (59.3%)
Median length of admission (days) (IQR)	7 (4–12)
**Clinical Features at Presentation**	
Initial median VA logMAR; Snellen equivalent, logMAR (IQR)	1.00; 20/200 (0.18–2.0)
ED ^1^ present (%)	139 (86.9%)
ED ^1^ size, median, mm^2^	4.3
Infiltrate present (%)	119 (74.4%)
Hypopyon present (%)	44 (27.5%)
Corneal thinning present (%)	61 (38.1%)
Final median VA; Snellen equivalent, logMAR (IQR)	0.48, ~20/40 (0.04–1.7)
**Risk Factors**	
Existing corneal disease	25 (15.6%)
Existing external eye disease	25 (15.6%)
Previous keratitis	24 (15%)
Contact lens use	20 (12.5%)
Foreign body	13 (8.1%)
Ocular Trauma	6 (3.8%)

^1^ ED = epithelial defect.

**Table 2 pathogens-12-00413-t002:** Table of organisms implicated in MK cases.

Name of Organism	N (%)
Gram positive	55/94 (58.5)
*Staphylococcus aureus* (*incl. MRSA* ^1^)	18 (15.1)
*Streptococcus pneumoniae*	10 (8.4)
*Propionibacterium acnes*	9 (7.6)
*Coagulase-negative staphylococci*	6 (5.0)
*Corynebacterium macginleyi*	4 (3.4)
Other Gram-positive: *Streptococcus sanguinis/milleri/dysgalactiae/viridans*, *Actinomyces* spp., *Bacillus* spp., *Staphylococcus lugdunensis*, *Enterococcus faecalis*	8 (6.7)
Gram Negative	39/94 (41.5)
*Pseudomonas aeruginosa*	17 (14.3)
*Moraxella* spp.	11 (9.2)
*Serratia marcescens*	4 (3.4)
Other Gram-negative: *Enterobacter cloacae*, *Proteus mirabillis*, *Escherichia coli*, *Pseudomonas oryzihabitans*, *Haemophilus influenzae*, *Morganella morganii*, *Stenotrophomonas maltophilia*	7 (5.9)
Fungi	
*Candida parapsilosis*, *Exserohilium*, *Aspergilus ruber*, *Fusarium* spp.	4 (3.4)
Virus	
Herpes Simplex Virus	19 (16.0)
Varicella Zoster Virus	2 (1.7)
Total	119 (100)

^1^ MRSA = Methicillin-Resistant Staphylococcus Aureus.

**Table 3 pathogens-12-00413-t003:** Bacterial keratitis species and sensitivity to antibiotics.

Organism	No. of Isolates	Sensitivity				
		Ciprofloxacin	Flucloxacillin	Penicillin *	Vancomycin	Gentamicin
Gram positive	37	3/3 (100%)	13/18 (72%)	15/25 (60%)	9/9 (100%)	1/1 (100%)
MSSA ^1^	13	1/1 (100%)	12/12 (100%)	1/5 (20%)	-	-
MRSA ^2^	5	-	0/5 (0%)	0/5 (0%)	5/5 (100%)	-
*Staphylococcus lugdunensis*	1	-	1/1 (100%)	-	-	-
*Streptococcus pneumoniae*	10	-	-	9/9 (100%)	-	-
*Streptococcus milleri*	1	-	-	1/1 (100%)	-	-
*Streptococcus dysgalactiae*	1	-	-	1/1 (100%)	-	-
*Corynebacterium macginleyi*	4	1/1 (100%)	-	1/2 (50%)	2/2 (100%)	-
*Bacillus* spp.	1	1/1 (100%)	-	1/1 (100%)	1/1 (100%)	1/1 (100%)
*Enterococcus faecalis*	1	-	-	1/1 (100%)	1/1 (100%)	-
Gram negative	38	26/27 (96%)	-	1/14 (7%)	-	25/26 (96%)
*Pseudonomas* spp.	18	17/17 (100%)	-	-	-	18/18 (100%)
*Moraxella* spp.	11	2/2 (100%)	-	0/7 (0%)	-	-
*Proteus mirabilis*	1	-	-	1/1 (100%)	-	1/1 (100%)
*Serratia marcescens*	4	4/4 (100%)	-	0/4 (0%)	-	4/4 (100%)
*Enterobacter cloacae*	1	1/1 (100%)	-	-	-	1/1 (100%)
*Escherichia coli*	1	1/1 (100%)	-	0/1 (0%)	-	1/1 (100%)
*Morganella morganii*	1	0/1 (0%)	-	0/1 (0%)	-	0/1 (0%)
*Haemophilus influenzae*	1	1/1 (100%)	-	-	-	-

^1^ MSSA = *Methicillin-Sensitive Staphylococcus Aureus*. ^2^ MRSA = *Methicillin-Resistant Staphylococcus Aureus.* * The sensitivity of organisms to cefazolin was extrapolated from testing penicillin.

**Table 4 pathogens-12-00413-t004:** Costs associated with MK at Liverpool Hospital from 2015–2020. 1 AUD = 0.68 USD (conversion performed on 13 December 2022).

	Inpatient Treatment Costs AUD (USD)		Outpatient Treatment Costs AUD (USD)	
	Median	Total	Median	Total
Admission	11,998 (8159)	2,069,386 (1,407,182)	-	-
A&E presentation	-	-	388 (264)	27,252 (18,531)
Follow Up Clinics	2079 (1414)	297,123 (202,044)	1485 (1010)	119,721 (81,410)
Wages Loss	2576 (1752)	318,941 (216,880)	966 (657)	76,314 (51,894)
Total Costs Direct	11,075 (7531)	1766,130 (1200,968)	1415 (962)	111,793 (76,019)
Total Costs Indirect	6810 (4631)	919,320 (625,138)	1419 (965)	111,495 (75,817)
Total Costs	18,003 (12,242)	2685,450 (1,826,106)	2839 (1931)	223,288 (151,836)

## Data Availability

The data presented in this study are available on request from the corresponding authors. The data are not publicly available due to patient confidentiality.

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
