# Peer review of "Epidemiology and Economic Cost Analysis of Microbial Keratitis from a Tertiary Referral Hospital in Australia"

_pathogens, 2023, doi:10.3390/pathogens12030413_

Round 1

Reviewer 1 Report

Dear Authors

I would like to stress out that I support the potential publication of this paper due to its scientific interest.

The paper is well written and interesting to read, however I see the following minor issues that should be resolved before publishing this paper:

Species names are not in italics, for example in line 23 Staphylococcus aureus (15.1%) and Pseudomonas aeruginosa

In the keywords add "Australia".

Add a sentence in the introduction section about the most frequently isolated agents and the reasons behind being the most isolated.

A significant part of the results is about the isolates/detection of pathogenic agents: bacteria, viruses and fungi. However, the methodology is poorly explained what makes it difficult to reproduce and should be more developed.

Author Response

Point 1: It was identified that microbial species names are not in italics in standing with normal conventions in scientific writing.

Response 1: All species names have been changed to italics.

Point 2: It was suggested that “Australia” be added to the keywords of the article.

Response 2: “Australia” was added to the keywords of the article.

Point 3: It was suggested that a sentence be added to the introduction about the most frequently isolated agents and the reasons behind why these were the most isolated.

Response 3: Two sentences have been added to the introduction from line 52 to line 56 explaining that Staphylococcus, Stretptococcus and Pseudomonas species are the most frequently isolated organisms in the overall literature, however, this can change depending on geographic location.

Point 4: A significant part of the results is about the isolates/detection of pathogenic agents: bacteria, viruses and fungi. However, it was identified that the methodology was poorly explained which makes it difficult to reproduce the study.

Response 4: The methods section has been updated in lines 90 to 94 and 96 to 102. This provides more detail on how cultures were collected and processed, subsequently improving the clarity with which others could reproduce this study.

Reviewer 2 Report

This study analyze the causative organisms of microbial keratitis, its presentation and economic burden from a tertiary referral hospital in Australia. The major concern is what is the novel findings in this study. There some minor concerns.

1.      Is there any difference economic burden between different pathogen such as Herpes Simplex, Staphylococcus aureus and Pseudomonas aeruginosa?

2.      The author would be better to discuss which factors influence admission or outpatient treatment (drug compliance, age, …) which effect economic burden.  

Author Response

Response to Reviewer 2 Comments

Point 1: There was concern that the novel findings in this study were not clear.

Response 1: The authors believe that this study is the most comprehensive cost analysis of the economic burden of microbial keratitis to date, with the reasons being listed in their conclusions in lines 327-337, and summarised below.

While this study is specific to an Australian tertiary referral hospital, it adds to the growing body of international literature, regarding the global burden of microbial keratitis Primarily, it highlights the disproportionate cost of admission and supports maximizing outpatient care, where safe to do so.

Additionally, it shows the prevailing pathogens responsible for microbial keratitis. This is important for monitoring trends in causative pathogens, as it can affect the efficacy of empiric therapy. Interstingly, our paper demonstrated the effective use of fluoroquinolone monotherapy for empiric treatment of disease. This indicates antimicrobial stewardship measures in Australia are successful in reducing resistance, and may have applications for reducing the rate of other multi-drug resistant organisms around the world.

Point 2: It was queried whether there was any difference in economic burden between different pathogens such as Herpes Simplex, Staphylococcus aureus and Pseudomonas aeruginosa?

Response 2: Statistical analysis was attempted to determine the economic burden of specific pathogens. However, there was not a large enough sample size for each individual pathogen to establish a statistically significant relationship to ecnomic burden. Without a sufficient sample size for individual pathogens, the authors decided they did not have sufficient data to comment on this aspect of economic burden and have subsequently ommited it from this paper.

Point 3: It was suggessted that the author would be better to discuss which factors influence admission or outpatient treatment (for example, drug compliance, age etc.) which effect economic burden. 

Response 3: Unfortunately, drug compliance was not consistently recorded in patient notes and was therefore unable to be assessed in relation to its effect on patient admission. However, the authors agree that factors that contribute to admission could be better explored and have subsequently made the changes in lines 194-202. There was a statistically significant association between the likelihood of admission and the age of the patient, the initial visual acuity and the size of the epithelial defect (a proxy for the severity of the microbial keratitis).

Reviewer 3 Report

Daley et al. reported the causative organisms of microbial keratitis, its presentation and economic burden from a tertiary referral hospital in Australia. They concluded that microbial keratitis represents a significant economic burden for eye related diseases and the key driving factor for cost is length of admission and minimizing the duration of admission, or opting for outpatient management would significantly reduce the cost. This manuscript was well prepared. The reviewer has one suggestion.

1.       Different hospitals have a variety of epidemiology and economic cost. The title of the manuscript should be: Epidemiology and economic cost analysis of microbial keratitis from a tertiary referral hospital in Australia.

Author Response

Response to Reviewer 3 Comments

Point 1: Different hospitals have a variety of epidemiology and economic cost. Subsequently, it was suggested that the title of the manuscript be changed to “Epidemiology and economic cost analysis of microbial keratitis from a tertiary referral hospital in Australia”.

Response 1: The authors agree with the reviewers comments and have updated the title of the article to, “Epidemiology and economic cost analysis of microbial keratitis from a tertiary referral hospital in Australia”.

Round 2

Reviewer 2 Report

It still requires collect more samples to get solid conclusions.

Author Response

Point 1: It was suggested that more samples were required for solid conclusions.

Response 1: The authors of this paper appreciate the reviewers’ ongoing considerations for our study. However, the authors disagree with the reviewers’ comment that the sample size is not sufficient. We feel a 5-year review of 160 cases provides the data required to give clinicians an adequate snapshot of the microbiological profile and economic burden of microbial keratitis. Additionally, the 5-year time period demonstrates a current cross-sectional view into antimicrobial resistance and economic expenditure for this specific time period. While a longer study may increase the sample size and show changes overtime, the most crucial conclusions that we are trying to convey, are the provision of an up-to-date microbiological profile, current resistance patterns and a contemporary economic cost analysis. Current trends in microorganism prevalence and effective antimicrobial treatment options provide the most clinically relevant information, which translates into effective empirical therapy. Our study is also consistent with other international literature that has attempted to conduct economic burden analysis of microbial keratitis across a 5 year span [1-3], with sample sizes similar to or less than this study [4-6].

References

  1. Cabrera-Aguas M, Khoo P, George R, Lahra M, Watson S. Antimicrobial resistance trends in bacterial keratitis over 5 years in Sydney, Australia. Clinical & experimental ophthalmology. 2019;48:183-91.
  2. Green M, Apel A, Stapleton F. Risk Factors and Causative Organisms in Microbial Keratitis. Cornea. 2008;27(1):22-7.
  3. Neumann M, Sjöstrand J. Central microbial keratitis in a Swedish city population. Acta Ophthalmologica. 1993;71(2):160-4.
  4. Moussa G, Hodson J, Virdee J, Penaloza C, Kigozi J, Rauz S. Calculating the economic burden of presumed microbial keratitis admissions at a tertiary referral centre in the UK. Eye (London). 2020;35(8):2146–54.
  5. Ashfaq H, Maganti N, Ballouz D, Feng Y, Woodward MA. Procedures, Visits, and Procedure Costs in the Management of Microbial Keratitis. Cornea. 2021;40(4):472-6.
  6. Khoo P, Cabrera-Aguas M, Robaei D, Lahra MM, Watson S. Microbial Keratitis and Ocular Surface Disease: A 5-Year Study of the Microbiology, Risk Factors and Clinical Outcomes in Sydney, Australia. Current eye research. 2019;44(11):1195-202.